Transcriptional landscape in rat intestines under hypobaric hypoxia

Tian Liuyang 1 2 3
Jia Zhilong 2 4
Xu Zhenguo 2 3
Shi Jinlong 2 3 4
Zhao XiaoJing xjingzhao@126.com 2 3
He Kunlun kunlunhe@plagh.org 2 3 4
1 School of Medicine, Nankai University , Tianjin , China
2 Beijing Key Laboratory of Chronic Heart Failure Precision Medicine, Medical Innovation Research Division of Chinese PLA General Hospital , Beijing , China
3 Military Translational Medicine Lab, Medical Innovation Research Division of Chinese PLA General Hospital , Beijing , China
4 Key Laboratory of Biomedical Engineering and Translational Medicine, Ministry of Industry and Information Technology, Medical Innovation Research Division of Chinese PLA General Hospital , Beijing , China
Mitsouras Katherine
Electronic publication date: 2021 Jul 29
Publication date: 2021
Volume: 9
Electronic Location ID: e11823
Received 2020 Dec 31; Accepted 2021 Jun 29
Copyright: ©2021 Tian et al.
Copyright year: 2021
Copyright holder: Tian et al.
License: This is an open access article distributed under the terms of the Creative Commons Attribution License, which permits unrestricted use, distribution, reproduction and adaptation in any medium and for any purpose provided that it is properly attributed. For attribution, the original author(s), title, publication source (PeerJ) and either DOI or URL of the article must be cited.
License URL: https://creativecommons.org/licenses/by/4.0/

Keywords: Intestine, Hypobaric hypoxia, Transcriptional analysis, Gene ontology, KEGG pathway

Funding: The National Key Research and Development Program of China 2017YFC0908403 National Natural Science Foundation of China 82001994 31701155 Chinese PLA General Hospital Clinical Research Support Fund 2018FC-WJFWZX-1-21 Chinese PLA General Hospital Youth Development Project QNC19058 This work was supported by the National Key Research and Development Program of China [grant number 2017YFC0908403], the National Natural Science Foundation of China [grant number 82001994,31701155], the Chinese PLA General Hospital Clinical Research Support Fund [grant number 2018FC-WJFWZX-1-21], and the Chinese PLA General Hospital Youth Development Project [grant number QNC19058]. The funders had no role in study design, data collection and analysis, decision to publish, or preparation of the manuscript.

==============================
Oxygen metabolism is closely related to the intestinal homeostasis environment, and the occurrence of many intestinal diseases is as a result of the destruction of oxygen gradients. The hypobaric hypoxic environment of the plateau can cause dysfunction of the intestine for humans, such as inflammation. The compensatory response of the small intestine cells to the harsh environment definitely changes their gene expression. How the small intestine cells response the hypobaric hypoxic environment is still unclear. We studied the rat small intestine under hypobaric hypoxic conditions to explore the transcriptional changes in rats under acute/chronic hypobaric hypoxic conditions. We randomly divided rats into three groups: normal control group (S), acute hypobaric hypoxia group, exposing to hypobaric hypoxic condition for 2 weeks (W2S) and chronic hypobaric hypoxia group, exposing to hypobaric hypoxic condition for 4 weeks (W4S). The RNA sequencing was performed on the small intestine tissues of the three groups of rats. The results of principal component analysis showed that the W4S and W2S groups were quite different from the control group. We identified a total of 636 differentially expressed genes, such as ATP binding cassette, Ace2 and Fabp. KEGG pathway analysis identified several metabolic and digestive pathways, such as PPAR signaling pathway, glycerolipid metabolism, fat metabolism, mineral absorption and vitamin metabolism. Cogena analysis found that up-regulation of digestive and metabolic functions began from the second week of high altitude exposure. Our study highlights the critical role of metabolic and digestive pathways of the intestine in response to the hypobaric hypoxic environment, provides new aspects for the molecular effects of hypobaric hypoxic environment on intestine, and raises further questions about between the lipid metabolism disorders and inflammation.

Introduction

Quick entry into the hypoxic environment of the medical plateau, defined as an area located 3000 m above sea level, can lead to serious damage to the organs of the body (Taino et al., 2019). In high-altitude areas, oxygen content and oxygen partial pressure decrease, forming a hypobaric hypoxic environment. Hypobaric hypoxia can lead to insufficient arterial blood oxygen content; this phenomenon is called systemic hypoxemia and causes cell and mitochondrial oxygen utilization disorders (Murray, 2016). The hypobaric hypoxic environment tends to increase the severity of inflammation, which can lead to brain damage (Li et al., 2017), pulmonary edema, and multiple organ dysfunction syndrome (MODS) which is the most complicated and difficult to cure (Swank & Deitch, 1996).

When entering high-altitude areas residents living at low altitudes will experience physiological reactions including changes in respiration and cardiovascular function, erythropoiesis factors, and gastrointestinal symptoms. The mechanism of the gastrointestinal symptoms of acute mountain sickness (AMS) caused by rising to high altitude is unclear (Fruehauf et al., 2020). Individuals with sufficient adaptability to high-altitude can maintain homeostasis under hypoxic conditions, but the metabolism and physiological functions will be affected, such as changes in energy metabolism in the heart and skeletal muscles. In other studies, the levels of Adenosine Triphosphate (ATP) and phosphocreatine (PCr) in skeletal muscle decreased with the altitude increases and this decrease continued over time. This indicates that decreasing ATP demand in high-altitude areas cannot meet the requirement of suppressing ATP supply (Hochachka et al., 1996).

In recent years, while excluding the symptoms of AMS, the incidence of gastrointestinal symptoms is extremely high, mainly manifested as abdominal pain, diarrhea, nausea, vomiting, and even hematochezia and melena for symptoms of gastrointestinal bleeding (Frühauf, 2014). Studies have shown that gastrointestinal mucosal barrier function damage caused by severe trauma, infection, hemorrhagic shock, and other factors can activate the inflammatory response pathway (Dempsey, Little & Cui, 2019). The intestine has essential biological functions at different oxygen concentrations (Sengupta, Ray & Chowdhury, 2014). Due to the existence of numerous gut microbiome, the intestine is in a hypoxic environment which maintains nutrient absorption, immune defense, and intestinal barrier functions (Singhal & Shah, 2020).

At present, various studies have shown that oxygen metabolism is closely related to the intestinal homeostasis environment and the occurrence of many intestinal diseases is a result of the destruction of oxygen gradients (Ramakrishnan & Shah, 2016). Under hypoxic conditions, cells adapt to the hypoxic environment by adjusting gene expression. Many metabolic changes are regulated by hypoxia inducible factors (HIFs). The encoding of enzymes regulated by HIFs involves glycolytic metabolism, lipid metabolism, nutrient absorption, peroxisomal metabolism, and mitochondrial function (Xie & Simon, 2017). Furthermore, it has been shown that hypoxia can increase leukocyte adhesion, stimulate mucosal tissues inflammation, and destroy the tissue barrier function (Deng et al., 2018; Saeedi et al., 2015; Seys et al., 2013). Regulating HIFs promote the expression of transcription factors in hypobaric conditions and through hypoxia, but how HIFs are regulated in hypobaric hypoxia and the downstream response is not clear (Wang et al., 2018). We conducted a more in-depth study through RNA-Seq analysis to explore the effects of 2 to 4 weeks of hypobaric hypoxic conditions on intestinal gene expression in in vivo studies.

Materials & Methods

Animals and treatment

Adult male Sprague Dawley (SD) rats were randomly divided into three groups with four rats in each group, but during modeling both the S and W2S groups lost one rat. According to the study by Qian Ni et al. (2014), we divided the experimental rats into three groups: the normal control group (S), the acute hypobaric hypoxia group (W2S) which was exposed to hypobaric hypoxic conditions for 2 weeks, and the chronic hypobaric hypoxia group (W4S) which was exposed to hypobaric hypoxic conditions for 4 weeks. The hypobaric hypoxic environment was constructed to simulate a 5,500-meter-high atmospheric environment using a FLYDWC50-1C hypobaric hypoxia cabin (Guizhou Fenglei Air Ordnance LTD, Guizhou, China). The rats were anesthetized with 10% chloral hydrate (0.4 ml/100g), the small intestine tissue was taken out, washed quickly, and stored in liquid nitrogen. The study was approved by the Animal Ethics Committee of the Chinese PLA General Hospital (2017-X13-05).

RNA extraction

The small intestine sample was chopped and mixed with liquid nitrogen to form a powder. Add Trizol reagent to dissociate nucleoprotein; add 0.3ml chloroform, shake for 15 s, and wait at room temperature for 2 min; 12000 rpm centrifuge (4 °C for 15 min), add 0.5 ml isopropanol and wait for 15 min at room temperature; 12000 rpm (4 °C for 15 min), aspirate the supernatant; add one mL 4 °C pre-cooled 75% ethanol to the RNA at the bottom; centrifuge (10,000 rpm, 5 min, 4 °C), and aspirate the supernatant; centrifuge (10,000 rpm, 4 °C, 5 min), aspirate the remaining liquid, and ventilate for 10 min; RNA concentration was determined in a Qubit2.0 Fluorometer (Life Technologies, Carlsbad, CA, USA) following the Qubit RNA Assay Kit operating instruction.

Transcriptome library construction and Sequence

Reverse transcription synthesis of the first strand cDNA, recombination into the second strand of DNA; then according to the above template in vitro transcription synthesis of cDNA, cDNA purification and reverse recording, purification, and quantification of the reverse transcription cDNA, and finally the library quality was assessed. The small intestine samples were sequenced with TruSeq PE Cluster Kit v3-cBot-HS (Illumina). All analyses are based on clean data.

Reference genome alignment

Reference genome was obtained from the relevant genome analysis website. The required gene annotation files were taken from the database as well. Using the HISAT2 v 2.0.4 comparison tool, the paired end readings were aligned with the reference genome (Langmead & Salzberg, 2012).

Differential expression analysis

The analysis of differentially expressed genes (DEGs) was finished through the Limma package in R language. Differentially expressed genes are based on the standard of —logFC—>1, P value<0.05. Genes with logFC >1 are defined as an up-regulated gene. Genes with LogFC <−1 are defined as a down-regulated gene.

Functional analysis and co-expression analysis

Gene ontology (GO) was implemented using the DEGs in three functional categories: biological process (BP), molecular function (MF), and cellular component (CC). The Kyoto Encyclopedia of Genes and Genomes (KEGG) pathway database was used to annotate DEGs on the pathway in the database for statistical analysis (Mao et al., 2005). The co-expression gene set analysis was finished using the Cogena software package (Jia et al., 2016).

qRT-PCR analysis

Total RNA was collected using the TRIzol reagent (TaKaRa, Japan) following the manufacturer’s instructions. Then RNA was reverse transcribed to cDNA using reverse transcriptase (Promega GoTaq qPCR Master Mix, China). qRT-PCR analysis was actualized by Bio-Rad CFX96 Real-Time PCR Detection System (Bio-Rad, Hercules, CA, USA).

Results

Differential gene expression analysis

Through differential gene expression analysis, we found that these three groups of samples had different responses to the hypobaric hypoxic environment. 636 differentially expressed genes were screened. These DEGs mainly involved ATP-binding cassette (Abc), ELOVL fatty acid elongase 2 (Elovl2), isocitrate dehydrogenase NADP(+) 1 (Idh1), apolipoprotein B (Apob), lipase C (Lipc), perilipin 1/2 (Plin1/ Plin2), fatty acid binding protein (Fabp), solute carrier family (Slc), angiotensin I converting enzyme 2 (Ace2), and membrane metallo-endopeptidase (Mme). The functions of these genes are associated with ATPase activity, fatty acid metabolic process, NADP metabolic process, cholesterol and lipid transport, thiamine (Vitamin B1) transmembrane transporter activity, and the renin-angiotensin system.

The hierarchical clustering of the DEGs is shown on the heat map in Fig. 1. W2S represented acute hypobaric hypoxia, while the W4S group represented chronic hypobaric hypoxia. The expression patterns of DEGs were vastly different from each other. After analysis, we found that the relationship between the W4S and the S groups were more closely related when compared with the W2S groups.

Figure 1 Heatmap of the DEGs.

Heatmap depicting the expression levels of differentially expressed genes (DEGs) among the Control (S1,S2,S4), acute hypobaric hypoxia (W2S1, W2S2, W2S3,) and chronic hypobaric hypoxia groups (W4S1, W4S2, W4S3, W4S4) based on the standard of —logFC— >1, P value <0. 05. 636 DEGs were found in the three groups. Genes with logFC > 1 are defined as up-regulated gene (red). Genes with LogFC < −1 are defined as a down-regulated gene (green).

There are 195/255 up-regulated genes in the W2S/W4S groups, respectively, compared with the S group. There are 120 up-regulated genes and 52 down-regulated genes in both the W2S and W4S groups compared with the S group and four genes up-regulated in the W2S but down-regulated in the W4S. Conversely, there are 19 genes down-regulated in the W2S but up-regulated in the W4S (Fig. 2, Table 1). Many of the up-regulated genes both in the W2S and the W4S groups, such as Tpsab1, Slc19a3, and LOC103694855, are associated with identical protein binding, transmembrane transport, and oxygen carrier activity. These up-regulated gene expressions may be associated with hypobaric hypoxia.

Figure 2 Differential gene expression in W2S and W4S.

The integration of upregulated DEGs and downregulated DEGs in groups W2S and W4S. 195/ 255 genes were up-regulated in group W2S/W4S respectively compared with group S. 120 genes were up-regulated and 52 genes were down-regulated both in groups W2S and W4S compared with group S. Four genes were up-regulated in group W2S but down-regulated in group W4S. Conversely, 19 genes were down-regulated in group W2S but up-regulated in group W4S.

Table 1 Differential gene expression in W2S and W4S.

List of up and down-regulated genes in the W2S and W4S groups.

Names	total	elements	
W2S up W4S down	4	Tff2 Plin1 Retn Spaca5	
W2S up W4S up	120	Acadm RGD1304770 Rbm11 Plekhs1 Nyap1 Nat8f4 Apoc2 Ces2 h Gpat3 Prss35 Bst1 Lin28a Gdf9 Usp2 LOC689065 Slc13a2 Epb41l3 Slc15a1 Trpm8 LOC100911440 Chdh Adamtsl5 Bdkrb1 Clec2e Ace2 Cyp2d5 Akr1c14 Amn Cyp2d4 B3gnt4 Asic3 Slc19a3 Pla2g12b Adam3a Slc43a2 Tmco2 S100a8 Tmem235 Fabp1 LOC498236 Fmo5 Iyd Alpp Mt1 Cyp2d3 Wfdc21 LOC500948 Dgat1 Apob Trim58 LOC103694874 Tmem37 LOC100909857 Cxcr2 Phlda2 Fabp2 Exoc3l4 Gucy2g Tmem120a Cyp4f39 Prr15 Anpep Slc5a11 Akr1b8 Mt2A Faim LOC691083 Apoa1 Hsd17b11 Slc12a3 Krt15 LOC100912163 Rorc Creb3l3 Gpt Tsku LOC498424 Bmp8b Mir194-2 Samd8 Adam32 Erich4 Prss12 Mfsd2a Fut7 Gcnt7 Pdzd3 Akr1b10 Cyp2d1 Reg1a Slc6a19 Tmem86a Slc7a7 Mme Gsdmd Tssk5 LOC689230 Moxd2 LOC100911949 Btnl5 Dusp9 Cyp2j4 Slc25a22 LOC103694864 Tmem45b Xpnpep2 Asz1 Bdnf LOC100910833 Mall LOC103694855 Tpsab1 Cyp3a9 Areg Pdzk1 Slc23a1 Acsl5 Slc30a10 Ahcyl2 Cdhr5	
W2S down W4S down	52	Ccdc173 C1ql3 Tmem158 Ppp1r42 LOC100911486 Megf6 Drd1 Dnah8 Pak7 St18 Ly6g6e Dll3 RGD1563354 Ly6g6d Aard LOC688801 Kcne5 Gba3 Gif Rag1 Nkx1-2 Gabra3 Sp7 UST4r Mzb1 Hmx2 Cacng6 Spata17 Lrrc3b Col19a1 Cldn24 Vsig4 Mtnr1a Lrrc15 Prss57 Nme5 Bean1 Nrsn2 Cfc1 Fam170b Fabp12 Krt81 Cyp26b1 Rgs22 LOC108348322 Kcng1 LOC690276 Tnfrsf17 Clca5 Klf8 Dppa3 Alkal1	
W2S down W4S up	19	Stra6 LOC685699 Mcpt4l1 Rln3 Kirrel2 F10 RGD1307603 Uox Clec7a Prrt4 Bbox1 LOC103690044 Mcpt4 Mybpc1 Oacyl Mcpt3 Lpar4 Htr5b Gjb5	
W2S up	71	Ly75 Ptgr1 Calcr LOC103691893 B3gnt3 Acox1 Akr1c2 LOC103690017 Adam22 Bche RGD1564463 Ddx43 Dapl1 Mdh2 Tnfaip2 Pabpc1l NEWGENE_620180 Alas2 Treml4 Plin2 Lgals4 Igsf21 Cry1 Upp1 Gsta4 Tldc2 Hpgd Ms4a4a Hprt1 Cd209f Adamts16 Otop1 Gpd1 Hba1 Nqo1 Aldh3a2 Il1r2 Sbp Idh1 RGD1561777 Mir223 Slc26a6 Cdkn2b Hmox1 Slc40a1 Hcn3 Abhd6 Sun3 Xpnpep1 Adtrp Gramd2 LOC289035 LOC100910401 Pklr Aldob Slc27a4 Irx2 Pbld1 Alox15 Naprt Hba2 Gckr Wfdc3 Knop1 Slc11a2 Sult1b1 Cyp27a1 LOC102548472 Rhd Gemin7l1 Reg4	
W2S down	152	Arhgap44 Eno4 Kcnd3 Slc2a12 Bend5 Syt2 Vegfd Chn1 Sarm1 Dusp27 Elovl2 Abcg4 Pla2g5 Scrn1 Dusp14l1 Mamdc2 Nrep LOC100909776 Enkur Adam23 Ldb2 Ccdc116 Tex52 Klhl32 Nup62cl Ppp2r2b Plp1 Gpr162 Slit2 Ndufa4l2 Lims2 Plekhb1 Grin2a Terc Calhm5 Atp6v0d2 Lrtm2 Snai3 Fmod Cspg5 RGD1359449 Ccdc3 Sh3tc2 Sema3e Cuzd1 Fam134b Sez6 Rfx2 Mustn1 Matn4 Syt15 Lrrc17 Hspb8 Moxd1 Trpc4 Dusp10 Cdh3 Adhfe1 Pdia5 Tmprss5 Dleu7 Ptprz1 Mapk4 RGD1311744 RGD1309651 Ctnna3 Igfals Cttnbp2 Prmt8 RGD1564899 Mreg Dusp1 Lpal2 Armcx6 Slc35f1 Sapcd1 LOC103690028 Chrna4 Unc5a Fxyd6 Syt17 Myh6 Fam198a Sorbs2 Pkhd1l1 Thbs2 Got1l1 Tnfrsf19 RGD1307537 Prelp Tnp2 Spata20 Efhd1 Cdh22 Snhg11 Atp2b4 B3gat1 Fam198b Otogl Plxnb3 Stum Zfp286a Nkx6-3 Nr1d1 Lrrc10b RGD1305645 Colca2 RGD1561102 Klhl38 Pygm Tspan18 Capn6 Tmem200a Hist1h1d Lypd6 Alpl Lgi4 Ptgfr Rab38 LOC102551716 Trim6 Cntn2 Patl2 Oas1g Itga7 Osbp2 Ak4 Abi3bp Rpl3l Ephb1 Kcnma1 Nlgn3 Arx Mx1 Chrdl1 Wnk2 Tppp3 Panx2 Dact3 Calhm4 LOC103689968 Ntm Tspan11 Ankrd6 Asb18 Adgrb2 Zkscan2 Slc9b2 Myh10 Tspan2 Arhgef25 Slc29a4	
W4S down	62	Tmco3 Calb1 Sall4 Rgn Esr1 Gpr81 Srl Coch Hsph1 Pcdhb12 Mir6333 Slpi Snai1 Hs3st6 RGD1565767 LOC100910275 F13b Gjc3 Lrrc55 Zfp2 Olr1588 Clip3 Slitrk6 Slc26a7 Ibsp Cebpe Cxcl2 Tcf21 Tp63 Tac4 Pcdh12 Slc27a5 Magee2 Plekhd1 Ccr3 RGD1563231 Clec4b2 Avpr1b Shisa2 Trim72 Rbm46 Pi15 Sim1 Efcab3 LOC102552988 Wfdc2 Kantr Trpa1 Ptch1 Cyp4f18 Ccr10 Fcer1a Nppc Lancl3 Cavin4 Ednra Ncan Il36b RGD1564801 Sstr2 Alox12 LOC100361018	
W4S up	116	Wdr63 Il13ra2 RGD1565212 Kiss1 Mir143 Ciart Hist1h4m Trpm6 Gpr17 Foxh1 Camk2b Padi3 Bcar3 Asb14 Car4 RGD1560608 Socs1 LOC103693823 Aplp1 Fam214b Slc52a3 Scn4b LOC100361556 Tubb1 C3 Ugcg Dnajb3 Mir1b Susd2 Neurod4 Rfk Ky Il10 Il4r Pex11a Hist2h2ab Mir142 Cnksr1 Lipc Arhgap26 Mir193a Acot11 Tmem211 RGD1310507 Ccnj Fam71f1 NEWGENE_1306399 Reg3b Abcb4 Abhd3 Olr1 RGD1564606 Acoxl Lrrc23 LOC102550375 Hist2h2ac Mir23b Gh1 Adgrf3 Noct RGD1309110 Iqcf1 Abcb1a Slc6a8 Abhd4 Gstm6l Rab24 Lamb3 Mxd3 Mir342 Tnip1 Slc2a9 Pomp Ido1 Tctex1d1 Sbspon Tmem232 Lpin2 Tnnc2 Rnf225 Mir339 Orm1 Cntn6 Tmem95 Sec16b Lca5l Ezr LOC367195 LOC103689920 Tbx18 Gpcpd1 B4galt6 Abcb1b Zc3h12a Steap4 LOC103689961 Csrnp1 Dgka Prr13 LOC100910838 Tsacc Unc5d LOC103693384 LOC100910656 Cmtm2a LOC100364769 Plk3 Grm3 Vpreb2 LOC500684 Matn1 Hamp Paqr7 Fam151a Cldn11 Lrrn4	

Further principal component analyses suggested similar results as the observation above (Fig. 3). The three groups are separated, but the members in the same group are clustered together; this indicated that the model construction was successful.

Figure 3 Principal component analysis of the samples in the three groups.

The principal component analysis for the DEGs among groups S, W2S and W4S. PC1 and PC2 are principal components 1 and 2, respectively.

Biological functional analysis of DEGs

To explore the impact of hypobaric hypoxia on gene function, we used Gene Ontology (GO) analysis. DEGs are classified into three functional categories. For biological processes, the GO analysis indicated that nutrient transport ontologies, especially anion, fatty, and acid were enriched. For molecular function, some transmembrane transporter ontologies, such as metal ion and active, were enriched. For cellular components, the DEGs were active in the plasma membrane (Fig. 4).

Figure 4 GO enrichment analysis of the DEGs.

DEGs are classified into three functional categories: (A) biological process. (B) molecular function. (C) cellular component. The Y-axis is the subordinate GO term of the three major categories of GO, and the x-axis is the number of genes under the term and the percentage of the total number of genes.

We used KEGG pathway enrichment analysis to further explore the biological functions of DEGs in response to hypobaric hypoxia. Several metabolic and digestive pathways, such as PPAR signaling pathway, glycerolipid metabolism, fat metabolism, mineral absorption, ferroptosis, and vitamin metabolism were identified (Fig. 5).

Figure 5 Pathway analysis of the DEGs.

The x-axis represents the Rich factor, while the y-axis represents the name of the pathway. The size of the dot represents the number of source genes in the pathway, and different colors indicate different FDR values.

Co-expression analysis

Co-expression analysis can identify genes with similar functions; in this co-expression analysis, we observed three patterns. Most of the genes in the first cluster were down-regulated in the 2nd week and up-regulated in the 4th week. In the second cluster, 126 genes were down-regulated until the 4th week. In the third cluster, 244 genes were up-regulated from the 2nd week until the 4th week (Fig. 6).

Figure 6 Heatmap of the co-expressed genes.

Three clusters (S, W2S,nW4S) determined by the k-means clustering method are shown. Most of these genes were down-regulated in the 2nd week and up-regulated in the 4th week in the first cluster. In the second cluster, most of these genes were down-regulated until the 4th week. In the third cluster, most of these genes were up-regulated from the 2nd week until the 4th week.

The Cogena Bioconductor software package was used for pathway analysis of each co-expression cluster. According to the manual, we selected three clusters for analysis and used the k-means clustering method. Interestingly, metabolic and digestive related pathways, such as PPAR and the renin angiotensin pathway, and fatty acid metabolism, are only enriched in the third cluster (Fig. 7). In the third cluster, we found that the metabolic and digestive pathways start to work in the 2nd week.

Figure 7 KEGG pathway analysis for coexpressed genes generated by cogena.

The color indicates the degree of statistical significance, and the enrichment score is the -log (q-value).

Gene transcription

The most changed genes, including the top 20 genes with increased expression and top 20 with decreased expression, have been listed (Fig. 8). The top five differentially expressed genes for each group, including the up-regulated genes and down-regulated genes, were detected through RT-PCR (Fig. 9). The trend of genes differentially expressed between each group is mostly matched with the transcriptional landscape.

Figure 8 The most changed genes.

The most changed genes including the top 20 genes with increased expression and the top 20 with decreased expression have been listed. (A) up-regulated genes in different groups. (B) down-regulated genes in different groups.

Figure 9 Relative expression levels of several DEGs for each group were detected by qRT-PCR.

(A) The up-regulated genes of the W2S group compared with the S group. (B) The down-regulated genes of the W2S group compared with the S group. (C) The up-regulated genes of the W4S group compared with the S group. (D) The down-regulated genes of the W4S group compared with the S group. (E) The up-regulated genes of the W4S group compared with the W2S group. (F) The down-regulated genes of the W4S group compared with the W2S group. * P < 0.05.

Discussion

The gastrointestinal tract is the key organ involved in the digestion of food and providing nutrients to the body for proper maintenance. However, the hypobaric hypoxic environment of the medical plateau can cause dysfunction of the gastrointestinal system. A previous study found that hypobaric hypoxia can effectively regulate immune/inflammatory processes and energy metabolism (Gangwar et al., 2020), and the degree of change has a clear relationship with time (Padhy et al., 2016). The high-altitude hypoxic environment destroys lipid metabolism and activates inflammatory processes, which causes abnormal metabolic function. Related studies have shown that the liver X receptor (LXR) plays an essential role in cholesterol, fatty acid, and glucose metabolism (Li & Glass, 2004). The physiological functions of LXRs have a close relationship with nuclear receptors (such as PPAR) (Hong & Tontonoz, 2014). The main role of the LXR pathway is based on energy homeostasis of lipid metabolism. In the high-altitude hypoxia model, the LXR pathway is strongly activated in the later stage (Tang et al., 2014). However, its mechanism of action in a hypoxic environment is still unclear, and the changes in the pathway may affect lipid metabolism. Several groups are currently conducting drug research and development based on LXR. Some studies have found that LXR activates macrophages through stearoyl-CoA desaturase (Wang, Kurdi-Haidar & Oram, 2004). The latest findings indicate that inflammation regulation is closely related to lipid metabolism. This study provides the first basis for investigating the changes in transcription and signaling pathways in rat small intestine tissues under acute and chronic hypobaric hypoxic conditions and found that the hypobaric hypotension environment affects fatty acid metabolism in the small intestine. These findings indicate that metabolic and digestive related pathways play a key role in the differences in the mechanism between hypobaric hypoxia and acute/chronic mountain sickness.

Studies have shown that oxygen metabolism is closely related to the intestinal homeostasis environment (Ramakrishnan & Shah, 2016). Under hypoxic conditions, many metabolic changes are regulated by hypoxia inducible factors (HIFs). The encoding of enzymes regulated by HIFs involves glycolytic metabolism, lipid metabolism, and nutrient absorption (Xie & Simon, 2017). Hypoxia can increase leukocyte adhesion, stimulate mucosal tissues inflammation, and destroy the tissue barrier function (Deng et al., 2018; Saeedi et al., 2015; Seys et al., 2013). By studying the changes of gene expression in small intestine tissues exposed to hypobaric hypoxia for 2 to 4 weeks, these results provide a basis for further research on the molecular pathogenesis of diseases under hypobaric hypoxic conditions. Through genome-wide transcription analysis, we found that hypoxia increased pro-inflammatory cytokines release, causing damage to the small intestine metabolism from the 2nd week. This study plays an important part in the research of the molecular mechanism of small intestine injury in high altitude areas and sets the stage for further genetic and functional research. There is a clear correlation between systemic inflammation and AMS (Boos et al., 2016; Li et al., 2015). A likely mechanism is that hypoxia changes cellular immunity and regulates the release of cytokines. Studies have shown that intermittent hypoxia has links with metabolic dysfunction (including abnormal lipid metabolism and insulin resistance). Intermittent hypoxia may induce sympathetic activation, increased inflammation, regulate hormone metabolism, and directly cause pancreatic β-cell damage to interfere with glucose metabolism (Drager, Jun & Polotsky, 2010). At the same time, in another study, scholars found that intermittent hypoxia conditions can increase the rate of lipid oxidation and suppress carbohydrate oxidation (Kelly & Basset, 2017). Previous reports have indicated that this severe intermittent hypoxia brings about a series of changes, such as low-grade inflammation, oxidative stress, and endoplasmic reticulum stress (Chacaroun et al., 2020). These adverse reactions are harmful to many systems, such as cardiovascular, respiratory, and metabolism, as well as cognition etc. (Chacaroun et al., 2020; Gabryelska et al., 2020; Gauda et al., 2020; Vermeulen et al., 2020).

In a previous study, Sweet R et al. found that hypoxia can up-regulate the expression of S100a8, the Retn gene, and Slc19a3 (Jiao, Wang & Zhang, 2019; Sweet, Paul & Zastre, 2010; Uchiyama et al., 2019). The transcriptional landscape and RT-PCR results of this study align with the trends of other researchers, indicating that this study will be a useful reference for future studies on intestinal diseases under hypobaric hypoxia. The angiotensin I converting enzyme 2 (Ace2) gene, which regulates systemic arterial blood pressure through the renin-angiotensin system, and the membrane metallo-endopeptidase (Mme) gene, that influences kidney development, are the DEGs found in this study, they play an important role in renin-angiotensin system and metabolic disorders related to the renin-angiotensin pathway were also observed. Revera M et al. found that during acute and chronic exposure to high altitude, the aortic pulse wave velocity increased significantly with altitude and the subendocardial oxygen supply/demand index significantly decreased (Revera et al., 2017). Acute exposure to hypobaric hypoxia can cause aortic stiffness and reduce subendocardial blood supply. A randomized clinical trial found that as the altitude increases, blood pressure will gradually increase, and will remain elevated for 3 weeks after entering the high-altitude area. Angiotensin receptor blockers can reduce blood pressure at 3400 m, but the effect is not apparent at 5400 m (Parati et al., 2014). A significant feature of hypoxic pulmonary hypertension is the remodeling of the pulmonary artery (Mikami et al., 1996). In another study by Mikami O et al. 1996, it was found that the angiotensin II (Ang II) receptors increased in the lung tissue of rats with hypoxic pulmonary hypertension, the Ang II level rose, and the renin-angiotensin system activated. Then, it induced hypertrophy and proliferation of aortic smooth muscle cells and cardiomyocytes (Parati et al., 2014).

Our study found that mineral absorption is overexpressed in a hypobaric hypoxic environment. Hypoxic environments and HIFs play an important role in the absorption of iron and minerals in the small intestine (Das et al., 2015; Das et al., 2020). The absorption of bones is affected by the hypobaric hypoxic environment (Guner et al., 2013) which causes abnormal absorption of minerals. This indicates that the hypobaric hypoxic environment will interfere with the metabolism of rats. This metabolic change may be the body’s adaptation to a hypobaric hypoxic environment.

This study has a few limitations including an insufficient number of subjects, which generated an accurate cut-off value for the diagnosis of AMS in rats. Also, since no blood samples were obtained in this experiment, we were unable to investigate plasma-related indicators.

Conclusion

This study found 636 DEGs associated with ATPase activity, fatty acid metabolic process, NADP metabolic process, cholesterol and lipid transport, thiamine (Vitamin B1) transmembrane transporter activity, and the renin-angiotensin system. KEGG pathway analysis found that in hypobaric hypoxic environments metabolic and digestive pathways, such as PPAR signaling pathway, mineral absorption, glycerolipid metabolism, fat digestion and absorption, ferroptosis, vitamin digestion and absorption, are highly enriched. Gene Ontology analysis revealed similar results. Cogena analysis discovered that up-regulation of digestive and metabolic functions began from the 2nd week of high-altitude exposure. This research provides new information regarding the molecular effects of the hypobaric hypoxic environment and raises new questions about lipid metabolism disorders, inflammation, and redox stress. Further study of the relationship between them is needed to explore the potential molecular mechanism of gastrointestinal regulation under hypobaric hypoxic environment.

Supplemental Information

Supplemental Information 1 Detailed information of DEGs between S, W2S and W4S groups, gene names, KEEG pathways and gene ontology

Click here for additional data file.

Supplemental Information 2 Author Checklist

Click here for additional data file.

Additional Information and Declarations

Competing Interests

Author Contributions

Animal Ethics

Data Availability

The authors declare there are no competing interests.

Liuyang Tian conceived and designed the experiments, performed the experiments, prepared figures and/or tables, authored or reviewed drafts of the paper, and approved the final draft.

Zhilong Jia analyzed the data, authored or reviewed drafts of the paper, and approved the final draft.

Zhenguo Xu performed the experiments, prepared figures and/or tables, and approved the final draft.

Jinlong Shi analyzed the data, prepared figures and/or tables, and approved the final draft.

XiaoJing Zhao and Kunlun He conceived and designed the experiments, authored or reviewed drafts of the paper, and approved the final draft.

The following information was supplied relating to ethical approvals (i.e., approving body and any reference numbers):

The Animal Ethics Committee of the Chinese PLA General Hospital provided full approval for this research (2017-X13-05).

The following information was supplied regarding data availability:

The data is available at NCBI SRA: BioProject PRJNA688382.

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
