# Peer review of "Transcriptional landscape in rat intestines under hypobaric hypoxia"

_PeerJ, doi:10.7717/peerj.11823_

## Round 0.1 · original submission · Major Revisions

· Academic Editor

Major Revisions

Your manuscript was considered interesting and valuable by the reviewers, but they raised some important comments that need to be addressed. Specifically, both reviewers suggested that you perform additional validation of the RNA Seq data, such as by RT-PCR and Western blotting. Additionally, one of the reviewers raised concerns about the validity of your model in eliciting hypobaric hypoxia and suggests that additional validation should be provided by assessing the transcript and protein levels of hypoxic markers.

Please, submit a detailed rebuttal which shows where and how you have taken all comments and suggestions into consideration. If you do not agree with some of the reviewers’ comments or suggestions, please explain why. Your rebuttal will be critical in making a final decision on your manuscript. Please, note also that your revised version may enter a new round of review by the same or by different reviewers. Therefore, I cannot guarantee that your manuscript will eventually be accepted.

Reviewer 1 ·

Basic reporting

There is some room for improvement in the English of the manuscript. I just listed a few places that need to be changed from the first page as below:

Abstract, line 20, “is a result of the destruction….”.
Line 23-24: “response to the hypobaric hypoxic…”
Introduction, “An area that locates 3000 meters above sea level is defined as…”

I would suggest the authors find an native English speaker to proofread the manuscript.

Experimental design

The authors mentioned in the Materials and Methods section that 2 rats were sacrificed during the experiment. One in group S and one in W4S. Can the authors explain what were the conditions caused the sacrifice? If they were related to the experimental settings, would it make sense to include their gene expression information into the analysis?

Validity of the findings

Have the authors confirmed some of the changes in gene expressions using RT-PCR or other techniques?

Additional comments

Can the authors provide a table listing the genes with the most changes (such as the top 20 genes with increased expression and top 20 with decreased expression), together with the folds of changes?

Reviewer 2 ·

Basic reporting

no commen

Experimental design

no commen

Validity of the findings

The authors need to verify the RNAseq results and provide experimental data, such as qRT-PCR and Western blots.

Additional comments

This study by Tian et al. reports the changes of the transcriptome in rat small intestine tissues under acute and chronic hypobaric hypoxic environment using RNA sequencing. The merit of this work is in the generation of transcriptional data and landscape of hypobaric hypoxia response in intestine tissues, which will be a useful database for future studies. However, there are numerous issues in this MS:
1. The hypobaric hypoxia environment is constructed as the simulated 5500-meter-high atmospheric environment using the hypobaric hypoxia cabin. The authors need to provide evidence to show the treatment can indeed extensively induce hypoxia response in treated rats, such as hypoxic markers on mRNA and protein levels.
2. In this study, how do the authors define acute and chronic hypobaric hypoxia (2- and 4-week treatment)?
3. Three groups of intestine samples, S, W2S, and W4S, were analyzed in this study. What is the transcriptional profile of each group, what are the common and unique DEGs of acute and chronic hypobaric hypoxic response compared to the control group, and what’s the difference between acute and chronic treatment?
4. The qRT-PCR and/or western blot need to be performed to verify the results of RNAseq in this study.
5. In the discussion, the authors need to focus on physiological and pathological characteristics of hypobaric hypoxia in intestines, analyze the correlation between transcriptome changes and hypoxic response discovered in this study.
6. There are spelling and grammar errors in this MS. The authors should seek the help of a native speaker who knows scientific writing to improve the writing and presentation.
7. For the figure legends, the authors need to provide detailed information of the figures, such as Figure 5 and Figure 6; Line 227, the position of citation is incorrect.

---

## Round 0.2 · accepted · Accept

· Academic Editor

Accept

Thank you for addressing their comments in a thorough fashion. As a result, your manuscript is much improved.

Reviewer 1 ·

Basic reporting

The authors addressed my previous concerns in the revision.

Experimental design

The authors addressed my previous concerns in the revision.

Validity of the findings

The authors addressed my previous concerns in the revision.

Reviewer 2 ·

Basic reporting

no comment.

Experimental design

No comment.

Validity of the findings

No comment.

Additional comments

The authors convinced me of the novelty, value, and validity of this study. There was no comment from my side.